# A Posteriori Dietary Patterns and Coronary Artery Disease in a Greek Case–Control Study

**DOI:** 10.3390/nu15224733

**Published:** 2023-11-09

**Authors:** Maria Dimitriou, Ioanna Panagiota Kalafati, Loukianos S. Rallidis, Genovefa Kolovou, George V. Dedoussis

**Affiliations:** 1Department of Nutritional Science and Dietetics, School of Health Sciences, University of Peloponnese, 24100 Kalamata, Greece; 2Department of Nutrition and Dietetics, School of Health and Education, Harokopio University of Athens, 17676 Athens, Greece; nkalafati@gmail.com (I.P.K.);; 3Second Department of Cardiology, Attikon Hospital, School of Medicine, National and Kapodistrian University of Athens, 11527 Athens, Greece; lrallidis@gmail.com; 4First Cardiology Department, Onassis Cardiac Surgery, 17674 Athens, Greece; genovefa@kolovou.com

**Keywords:** coronary artery disease, cardiovascular disease, dietary patterns, Western-type dietary pattern, factor analysis

## Abstract

Introduction: Diet is one of the most important modifiable risk factors associated with cardiovascular health (CH). Research identifying dietary patterns (DPs) through data-driven analysis and reporting associations between DPs and coronary artery disease (CAD) outcomes is rather limited. Objective: The aim of the present report was to generate DPs through factor analysis (FA) and to examine their association with CAD risk. Methods: Participants (*n* = 1017) consisted of cases diagnosed with CAD (*n* = 356) and controls (*n* = 661) drawn from the THISEAS study. Demographic, anthropometric and lifestyle data were collected. Dietary components were generated through FA. Logistic regression analysis was performed to estimate CAD relative risks. Results: FA generated seven dietary components, explaining 53.5% of the total variation in intake. The Western-type DP showed a modest significant association with CAD risk, after controlling for confounders (OR = 1.20; 95% CI = 1.09–1.32, *p* < 0.001). The vegetarian-type DP was not significantly associated with the likelihood of CAD (OR = 0.95; 95% CI = 0.84–1.04, *p* = 0.259). Discussion: The Western-type DP was positively associated with CAD risk and the odds were further increased after controlling for confounders. This finding is in concordance with previously reported positive associations between Western patterns and CAD risk. Limited data exist regarding a posteriori DPs and their effect on CAD risk.

## 1. Introduction

Global statistics place cardiovascular disease (CVD) mortality first, since CVD deaths represent 32% of all deaths worldwide and 38% of premature deaths due to noncommunicable diseases (NCDs). Coronary artery disease (CAD) is the most common type of CVDs and it is estimated that the number of people that will die due to CAD will rise up to 23.6 million by the year 2030 [1].

Low and moderate levels of cardiovascular health (CH) are attributable to the majority of CVD events in the United States, according to the American Heart Association (AHA)’s updated statistics report [2]. Therefore, premature mortality could be reduced or even avoided when focusing on CH level improvements.

Diet is one of the most important CAD risk factors, depicts health behavior and can modify CAD risk and overall CVD risk [3,4]. According to AHA, diet is among the seven approaches for better CH [2]. Research investigating the impact of specific dietary nutrients of foods on CAD may highlight their protective effects against CAD and is still ongoing [5,6,7,8,9]. However, the nutrients and bioactive chemicals of food items are inter-correlated and research for associations between a single nutrient and a chronic disease may underestimate the impact of the overall diet on health outcomes [10]. In recognition of the synergy of nutrients, research was directed towards the study of dietary patterns (DPs) in the prevention or treatment of disease [11,12]. Therefore, apart from focusing on the protective potential of individual nutrients, it is also important to study the impact of DPs on the disease. A DP conceptualizes the nutritional intake, the quality and the variety of the overall diet.

Although there is supportive evidence regarding the protective effect of the Mediterranean-type diet on CAD risk, data regarding other known DPs such as, vegetarian-type, and Western-type diets remain limited. Furthermore, research identifying DPs through data-driven analysis and reporting associations between DPs and CAD outcomes is scarce, being even more scarce in the Greek population, and the results remain inconclusive. In addition, the posteriori approach is independent of current nutrition-model knowledge and is an important tool with which to identify novel DPs that may substantially increase or decrease CAD risk.

In order to bridge the gap and provide more evidence to the current literature regarding DPs on disease risk, the aim of the present report was to identify DPs through data-driven analysis and to evaluate their association with CAD risk.

## 2. Materials and Methods

### 2.1. Study Design and Population

Details regarding the materials and methods used for the study population, along with demographic, anthropometric, clinical and lifestyle assessments, have been previously published [13]. The study population comprised up to 1017 subjects of Greek origin drawn from the THISEAS database, constituting a case–control study.

Cases were coronary patients presenting with acute coronary syndrome or stable CAD defined as >50% stenosis in at least one of the three main coronary vessels. All patients had undergone coronary angiography. Controls were individuals free of CAD. Exclusion criteria from both study groups were the presence of renal or hepatic disease. Subjects from the THISEAS database with incomplete/missing dietary data and missing data regarding other parameters tested were excluded from current analyses.

Therefore, the analysis of the present report was restricted to 356 cases diagnosed with first-time CAD at the time of recruitment and 661 controls, depending on the dietary data availability of the cohort. The study protocol was approved by the Ethics Committee of Harokopio University of Athens. The flow chart of the present study is depicted in Figure 1.

### 2.2. Demographic and Lifestyle Characteristics

All participants were interviewed regarding their origins to ensure their Greek ancestry. Data regarding educational status were collected and in the present work educational status was measured via years of schooling.

Physical activity level (PAL) was assessed through the Harokopio Physical Activity Questionnaire (HAPAQ), which evaluates the frequency, duration and intensity of occupational, household and leisure time activities [14]. Physical activity (PA) adoption was assessed as a categorical variable categorizing the participants into two groups, based on whether they reported leisure time activities in a regular basis or not. Volunteers that reported no leisure time activities were categorized as physically inactive.

Participants were classified into current, never or former smokers. Information regarding the average daily number of cigarettes, the duration of smoking and time of cessation was also obtained. Current smokers were defined as those who smoked at least one cigarette per day, non-smokers those who had never smoked in their life and former smokers those who had stopped smoking for at least six months. In the present analysis, former smokers were combined together into one group with never smokers.

### 2.3. Anthropometric Measurements

Body weight (BW) and height (Ht) were measured in all participants, who were wearing light clothing, without shoes. Weight was measured to the nearest 0.5 kg using a leveled platform scale. Height was measured to the nearest 0.5 cm using a wall-mounted stadiometer. Body mass index (BMI) was computed as weight (kg)/height^2^ (m) (Quetelet’s equation):BMI = BW (kg)/[Ht (m)]^2^

### 2.4. Dietary Assessment

Nutritional information was recorded through a 172-item picture-sort food frequency questionnaire (FFQ). Dietary data were manually entered into an Excel spreadsheet database that translated the queried foods and beverages into food group equivalents. Regarding combinations of individual foods into one food item, the researcher referred to the ingredients, nutrient information and recipe from reference lists. In order to calculate portion sizes, the dietary guidelines for adults in Greece were used [15]. In total, 26 food groups were estimated (Appendix A). For analysis, the number of food groups was further narrowed down to 21 by including two similar food groups in one (e.g., dairy, full fat, and cheese, full fat).

### 2.5. Statistical Analysis

Continuous variables are presented as mean values and SD, while categorical variables are presented as relative frequencies. Differences between categorical variables and groups of the study were assessed using the χ^2^ test. P–P plots were applied to assess the normality of the distribution of the continuous variables. Student’s *t* test or the Mann–Whitney test was applied to evaluate differences in continuous variables between the two study groups.

The factor analysis (FA) technique was used to identify and generate DPs. Data preparation was completed prior to performing FA. This step included the quality control of the dataset by checking for outliers regarding consumption, removing missing data and ensuring continuous variables. Given that most studies have demonstrated a U-shaped relationship between CAD risk and coffee or alcohol consumption, alcohol and coffee were dropped from further analysis [16,17]. In total, 19 food groups were coded as servings per day and underwent FA. Exploratory FA was carried out to evaluate validity, disclose underlying structures and reduce the number of variables. FA was chosen as the extraction method using orthogonal rotation (Varimax rotation) in order to generate non-correlated components (namely, non-correlated DPs). The food variables that were highly correlated showed factor loadings (correlation coefficients) greater that |0.4|. The cut-off point for Eigen values was greater than 1.0 [18].

Logistic regression models (unadjusted and adjusted for major confounders) were used in order to estimate the relative risks of developing CAD via the calculation of ORs and their corresponding 95% confidence interval. Model 2 logistic regression was adjusted for main covariates, namely age, sex and BMI. In order to control for more covariates (beyond Model 2 covariates) related to the dependent variable and to alleviate concerns regarding a loss of study power due to multiple testing, we tested the selected set of covariates one at a time. Covariates retained as significant were the presence of hypertension, diabetes mellitus, dyslipidemia and current smoking. Physical inactivity and years of education that changed the effect size by less than 10% and were non-significant were removed. Therefore, we concluded on using Model 3 adjusted for age, sex, BMI, presence of hypertension, diabetes mellitus, dyslipidemia and current smoking.

Analyses were based on 2-sided tests, while statistical significance was set at *p* ≤ 0.05. The statistical software package IBM SPSS Statistics 21.0 (SPSS Inc., Frisco, TX, USA) was used for all statistical calculations, where appropriate.

## 3. Results

Table 1 presents the descriptive characteristics of the study (namely, the demographic, lifestyle and clinical characteristics). The two study groups significantly differed regarding age; specifically, cases with CAD were older compared to controls (*p* < 0.001). A higher proportion of males comprised the case groups, compared with that in the control group (*p* < 0.001). As expected, the prevalence of arterial hypertension, hypercholesterolemia, type 2 diabetes mellitus (T2DM), physical inactivity and cigarette smoking was higher in cases compared to that in controls.

Table 2 depicts the score coefficients (factor loadings) derived from FA. Absolute values greater than 0.4 indicate that the food variables are highly correlated and contribute more to the development of a dietary component. In this report, seven components were generated from the initial food groups, and explained 53.5% of the total variation in intake. Specifically, the derived components from the analysis were as follows:(1)Component 1, a Western-type pattern, which included red meat, processed meat, fried potatoes and fast foods;(2)Component 2, a vegetarian-type pattern, which is mainly characterized by vegetables, legumes and potatoes (boiled, baked or smashed);(3)Component 3, a starch pattern, which was loaded with refined and unrefined starch (with the unrefined starch prevailing)l(4)Component 4, a pattern that was characterized by the consumption of poultry, fish and seafood;(5)Component 5, a pattern that included dairy and eggs,(6)Component 6, a binge eating-type pattern that included the intake of sweets and nuts;(7)Component 7, a pattern that included soft drinks and fruit drinks. Fruits and fresh fruit juice were not loaded.

**Table 2 nutrients-15-04733-t002:** Loadings from principal component analysis regarding food groups consumed by the participants from the THISEAS study.

	Component ^a^						
	1	2	3	4	5	6	7
Red meat	**0.584**	0.300	−0.117	0.043	0.087	0.033	−0.122
Processed meat	**0.671**	−0.076	0.097	−0.006	0.031	0.066	0.082
Potatoes, fried	**0.549**	0.135	−0.175	0.007	0.189	−0.189	0.039
Fast foods	**0.630**	−0.083	−0.165	0.037	0.152	0.149	0.173
Vegetables	−0.105	**0.625**	0.164	0.268	0.089	−0.097	0.198
Legumes	−0.036	**0.666**	−0.147	−0.010	−0.046	0.297	−0.145
Potatoes, boiled/baked/smashed	0.208	**0.615**	−0.054	−0.034	0.082	−0.031	−0.010
Refined starch	0.139	0.256	**−0.645**	−0.057	0.056	0.056	0.124
Unrefined starch	−0.030	0.096	**0.813**	0.020	0.054	−0.041	−0.092
Fish	−0.007	0.208	0.031	**0.606**	−0.048	−0.004	−0.168
Seafood	0.204	−0.108	−0.059	**0.688**	0.065	−0.028	0.135
Poultry	*0.537*	0.082	0.098	**0.431**	−0.331	−0.026	−0.099
Dairy, full fat	0.096	0.298	0.003	0.089	**0.565**	−0.193	0.041
Dairy, semi/non fat	−0.033	−0.020	*0.443*	−0.201	**−0.440**	−0.214	0.292
Eggs	0.121	−0.026	0.019	−0.084	**0.624**	0.193	−0.011
Sweets	0.214	−0.038	0.032	0.132	0.336	**0.498**	0.292
Nuts	0.024	0.095	0.024	−0.029	−0.029	**0.739**	−0.070
Soft drinks	0.180	0.112	−0.023	−0.255	−0.058	−0.225	**0.701**
Fruit drinks	−0.02	0.081	−0.158	0.224	−0.037	0.184	**0.590**
Fruits	−0.219	0.243	0.0201	0.366	−0.020	0.279	0.215

Numbers in bold indicate loadings with an absolute value of >0.4 (a higher correlation of the food group with the component); total %variance explained equals 53.5. ^a^ Component description: Component 1 = a Western-type pattern; Component 2 = a vegetarian-type pattern; Component 3 = a starch-type pattern; Component 4 = a pattern that is mainly characterized by the consumption of poultry, seafood and fish; Component 5 = a pattern that is mainly characterized by the consumption of dairy and eggs; Component 6 = a binge eating-type pattern that is mainly characterized by the consumption of sweets and nuts; Component 7 = a pattern that is mainly characterized by the consumption of soft drinks and fruit drinks.

In order to evaluate the associations between each extracted dietary component and CAD risk, logistic regression models were performed without adjustments (Model 1) or after controlling for main covariates (Model 2, adj. for age, sex and BMI). In addition, Model 3 was adjusted for more covariates (Model 2 + presence of arterial hypertension, dyslipidemia, diabetes mellitus and current smoking). Table 3 depicts the results from logistic regression, which evaluated the association between each dietary component and CAD likelihood. The unadjusted regression showed that Component 1 (OR = 1.10; 95% CI = 1.01–1.10, *p* = 0.034), Component 4 (OR = 1.03; 95% CI = 1.02–1.04, *p* = 0.000) and Component 5 (OR = 1.09; 95% CI = 1.03–1.16, *p* = 0.003) were positively associated with CAD risk. On the other hand, Component 3, Component 4 and Component 7 were inversely associated with CAD risk. After adjusting for Models 2 and 3, only the association of Component 1 (Western-type diet pattern) remained significant (Model 2: OR = 1.20; 95% CI = 1.09–1.32, *p* < 0.001) (Model 3: OR = 1.13; 95% CI = 1.02–1.24, *p* < 0.017). Model 2 revealed a modest effect of Component 1 on CAD risk. The effect size was attenuated after Model 3 analysis, although it remained significant. The positively associated Component 5, along with the inversely associated Components 3, 4 and 7 lost their significance in adjusted analyses. In addition, Component 6 was inversely associated with CAD risk (Model 1 and 2) but lost significance in Model 3 analysis. Component 2 did not demonstrate significant associations with the likelihood of CAD, in both unadjusted and adjusted analyses.

Furthermore, in an attempt to apply multiple regression analysis to model the associations between DPs, dependent variables and CAD (total cholesterol, low-density-lipoprotein cholesterol, triglycerides {TG} and systolic blood pressure) in the pooled sample or in controls did not reveal significant results.

## 4. Discussion

The present report attempted to create posteriori DPs and assess their associations with the likelihood of having CAD. Dietary components were identified from food groups that underwent the FA, a data-driven statistical method. In total, seven components (DPs) were generated.

The first component could be described as a Western-type DP and depicted an unhealthy pattern; it was mainly characterized by the consumption of red meat, processed meat, fried potatoes and fast foods. This pattern was positively associated with CAD risk. Although the odds were further increased after controlling for main confounders (Model 2) compared to those under the unadjusted analysis (Model 1), the effect was attenuated when controlling for more confounding variables (Model 3). However, in all models the association of the pattern remained significant. This finding is in concordance with the previously reported positive association between Western-type patterns and CAD in studies with high methodological quality [19]. Western-type DPs are mainly characterized by saturated fatty acid intake, and this type of fat has been implicated as a CAD risk factor. In another Greek study, a pattern characterized by meat intake and meat products has been associated with higher waist circumference and lower levels of high-density-lipoprotein cholesterol (HDL-C) [20]. Nowadays, there is evidence in the literature indicating that the Western-type diet is associated with CVD, as well as other NCDs (such as obesity, T2DM and cancer) due to their regulation of the gut microbiota–immune system interaction [21]. In the present report, the Western-type pattern that was generated from the analysis contained ultra-processed foods (meat products and fast food) which are associated with an increased prevalence of CVD [22,23,24].

On the other hand, Component 2, which is a vegetarian-type DP, is mainly characterized by the consumption of vegetables, legumes and potatoes (fried potatoes were excluded) and depicts a healthy DP. Vegetables and legumes are usually components of prudent patterns, which are inversely associated with CAD risk [19]. In our findings, this component showed an expected directional effect on CAD risk, although it was not significant. In general, there is some evidence in the literature suggesting a protective effect of vegetarian type patterns on primary prevention [25]. Vegetarian-type diets seem to have a protective effect against CVD risk and improve overall CH. However, more consistent associations regarding cardiovascular benefits need to be observed [26].

The inverse association of the seventh pattern (sweets and nuts) (Model 1 and 2) is in line with the results of a case–control Norwegian study. The latter demonstrated an inverse association of sweets with myocardial infarction [27]. The authors mention that although this was an unexpected outcome, this food group contained food items, such as, nuts, almond paste and chocolate, which have been associated with reduced CAD risk. Similarly, in our report, this component also included nuts; in addition, among other sweets, almond chocolate and dark chocolate were also included. We cannot obviate the possibility that the latter food items affected the direction of the association with CAD risk. This association lost significance after Model 3 analysis.

It has been demonstrated that diets characterized by foods with high glycemic index scores have been associated with high TG and low HDL-C levels [28]. In addition, beverages with added sugar are associated with CAD risk [29]. However, in this report, soft drinks and fruit drinks (Component 7) did not reveal associations with CAD risk, after confounding. The third, fourth and fifth components also lacked significant associations, after confounding.

Potential limitations of this report are the recall bias of food intake, which may have resulted in underreported or overreported dietary intakes. Many cases had received dietary advice by the time of the interview, so we cannot rule out the possibility that this may have influenced their diet report towards the intake of favorable foods indicated in the dietary advice. This is a case–control investigation that cannot support causality. In addition, the two study groups significantly differed in gender proportions. Although logistic regression Model 2 was adjusted for the sex variable, this does not preclude the possibility of a sex misbalance affecting the outcome of some results. FA as a technique also has some limitations, since the extracted components are based on subjective decisions.

We assume that some of the non-significant findings may well have been significant if the sample size was larger and had a larger discriminatory power.

Despite the limitations of this report, we assume that these results are noteworthy, given the limited evidence in the literature identifying DPs through FA and examining DPs for CAD risk. Most studies investigate possible associations between a priori DPs and cardiovascular disease risk [30]. Research in the Greek population consistently reveals a protective effect of the Mediterranean diet on CAD risk, examined as an a priori DP [31,32]. However, research on posteriori DPs and CAD outcomes is scarce, if there is any, meaning we are unable to compare our results with others. Therefore, more research is needed in this direction that could eventually identify the optimal diet for disease prevention.

## 5. Conclusions

Regarding an important environmental parameter, diet, research using DPs highlights the importance of food diversity in health outcomes. Although dietary assessment is approached using different methods, many studies have consistent results regarding DPs and CAD. These results are important and may contribute to the implementation of programs and services supporting overall healthy patterns against disease prevention and development instead of the avoidance of certain foods or nutrients. Policy health promotion can encourage the improvement of eating behaviors at an individual and population level.

## Figures and Tables

**Figure 1 nutrients-15-04733-f001:**
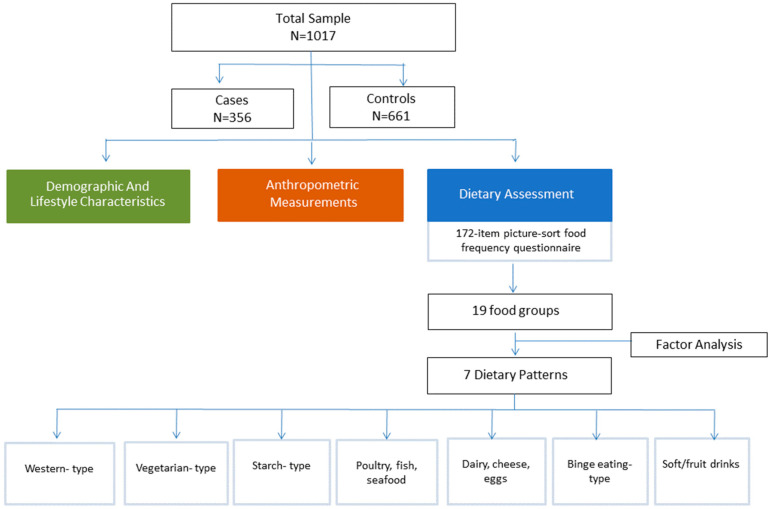
Study flow chart.

**Table 1 nutrients-15-04733-t001:** Descriptive characteristics of the participants.

	Controls (N = 661)		Cases (N = 356)		
	Mean or Frequency	±SD *	Mean or Frequency	±SD	*p*-Value
Demographic & Lifestyle characteristics					
Age (years)	54.1	±14.1	62.5	±10.1	<0.001
Male sex (%)	49.5		82.1		0.000
Years of education	12.3	±4.6	11.5	±4.9	0.007
	**Relative Frequency (%)**		**Relative Frequency (%)**		***p*-Value**
Physical inactivity	79.5%		90.9%		0.000
Current smokers	26.4%		46.7%		0.000
Clinical characteristics					
	**Mean**	**±SD**	**Mean**	**±SD**	***p*-Value**
Body mass index	28.4	±4.9	27.8	±3.8	0.040
Systolic blood pressure (mmHg)	134	±18	134	±20	0.944
Diastolic blood pressure (mmHg)	80	±11	80	±13	0.658
Total cholesterol (mg/dL)	210	±39	192	±48	0.000
Low-density lipoprotein cholesterol (mg/dL)	133	±35	123	±42	0.000
Triglyceride (mg/dL)	114	±64	148	±103	0.000
Blood glucose (mg/dL)	98	±23	113	±35	0.000
	**Relative Frequency (%)**		**Relative Frequency (%)**		***p*-Value**
Prevalence of hypertension	47.5%		90.3%		0.000
Use of antihypertensive medication	29.0%		85.1%		0.000
Prevalence of hypercholesterolemia	69.5%		88.4%		0.000
Use of lipid lowering medication	21.5%		79.4%		0.000
Prevalence of diabetes mellitus	10.6%		35.0%		0.000
Use of anti-diabetic medication	5.7%		21.8%		0.000

* SD = standard deviation.

**Table 3 nutrients-15-04733-t003:** Results from logistic regression, which evaluated the association between dietary components and the likelihood of having coronary artery disease.

	Odds Ratio	95% CI	*p*-Value
Component 1: ^a^ Western-type dietary pattern (DP)			
* Model 1	1.10	1.01–1.10	0.034
** Model 2	1.20	1.09–1.32	<0.001
*** Model 3	1.13	1.02–1.24	0.017
Component 2: ^b^ Vegetarian-type DP			
Model 1	0.97	0.90–1.05	0.48
Model 2	0.95	0.84–1.04	0.26
Model 3	0.94	0.85–1.04	0.22
Component 3: ^c^ Starch-type DP			
Model 1	0.95	0.91–0.99	0.007
Model 2	0.98	0.94–1.03	0.45
Model 3	1.00	0.95–1.05	0.97
Component 4: Poultry, fish and seafood DP			
Model 1	0.75	0.61–0.92	0.005
Model 2	0.85	0.67–1.07	0.15
Model 3	0.80	0.61–1.04	0.10
Component 5: Dairy, cheese and eggs DP			
Model 1	1.09	1.03–1.16	0.003
Model 2	1.06	1.00–1.14	0.06
Model 3	1.06	0.98–1.15	0.12
Component 6: ^d^ Binge eating-type DP			
Model 1	0.79	0.71–0.89	<0.001
Model 2	0.84	0.75–0.95	0.005
Model 3	0.88	0.77–1.01	0.07
Component 7: ^e^ Soft/fruit drinks DP			
Model 1	0.66	0.52–0.83	0.001
Model 2	0.77	0.77–1.00	0.052
Model 3	0.83	0.62–1.12	0.23

* No adjustments; ** adjustments = age, sex and body mass index; *** adjustments = age, sex, body mass index, presence of arterial hypertension, dyslipidemia, diabetes mellitus and current smoking. ^a^ A pattern mainly characterized by the consumption of meat, processed meat, fast foods, fried potatoes and fast-food; ^b^ a pattern mainly characterized by the consumption of legumes, vegetables and potatoes; ^c^ a pattern mainly characterized by the consumption of unrefined starch; ^d^ a pattern mainly characterized by the consumption of sweets and nuts; ^e^ a pattern mainly characterized by the consumption soft drinks and fruit drinks.

## Data Availability

Data are available on request from chief investigators due to ethical and privacy restrictions.

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
