# Peer review of "A Posteriori Dietary Patterns and Coronary Artery Disease in a Greek Case–Control Study"

_nutrients, 2023, doi:10.3390/nu15224733_

Round 1

Reviewer 1 Report

Comments and Suggestions for Authors

This manuscript by Dimitriou et al. describes the results of a statistical analysis of dietary data related to CAD risk in a subset of cases and controls of the THISEAS Study. The statistical technique of factor analysis is applied in the search for dietary patterns that may be associated with cardiovascular risk.

The report agrees in its results and conclusions with many other studies, but with the added value that dietary patterns emerge from the analysis of the data instead of being stablished a priori.

However, some points should be addressed by the authors in order to clarify the presented data and the conclusions they extract from.

·         Gender distribution is not shown in the descriptive Table 1. However, it is considered as a confounder in regression analysis and, accordingly, adjusted for (Table 3). This information should be shown together with age and, if a misbalance exists between groups could have an impact in the results and conclusions.

·         Regarding logistic regression analysis, the description of the procedure followed seems to be incomplete in the Materials and Methods section. According to Table 3 footnotes, the adjusted models take into account only Age, Sex, and BMI, but no justification to include these variables (and no others that are shown to be different between Cases and Controls –physical activity, smoking status, for example) is given. How did the confounding variables were chosen for the adjusted model?

   Have the authors verified all data requirements to apply factor analysis appropriately? This information should be included in the Materials and Methods section.

Reviewer 2 Report

Comments and Suggestions for Authors

GENERAL COMMENTS

This is an interesting and timely topic. The study has the strength of focusing on dietary patterns instead of o single foods as well as the big simple size with over 1000 individuals. The data are derived from a Greek case-control cohort evaluating CAD. Seven dietary components were derived including the Western type diiiet and the Vegetarian one. Some of the findings are of confirmatory nature but nonetheless of interest.

Some suggestions are provided to further improve the manuscript.

Abstract: indicate n value and p values.

Page 2, line 53: formulate the specific working hypothesis; what did you expect and why. Subsequently you can indicate your aims.

Page 2, line 75: should be “… physically inactive.”?

Material/Methods, Page 2: leave a space between the number and the unit.

Material/Methods, Page 3, line 111: replace “generated” by “generate”.

Results, Page 2 Lines 121-124: indicate that cases were significantly older than controls.

Results, Table 1: do not use decimals for mean and SD data of SBP, DBP, cholesterol triglycerides, glucose.

Results, Table 2: do you have information on alcohol consumption?

Discussion, page 7, line 192/193: the expression “Last but not least” is a bit colloquial.
